# Using Functionally Redundant Inertial Measurement Units to Increase Reliability and Ensure Fault Tolerance

**Ivan M. Kuznetsov \*, Konstantin K. Veremeenko, Maxim V. Zharkov**  **and Andrey N. Pronkin**

Flight-Navigation and Information-Measuring Systems Department, Moscow Aviation Institute (National Research University), 125993 Moscow, Russia; nio3@mai.ru (K.K.V.); mv_zharkov@mai.ru (M.V.Z.); an_pronkin@mai.ru (A.N.P.)

\* Correspondence: kuznetsovim@mai.ru

**Abstract:** This paper aims to assess the possibility of using functionally redundant inertial units to solve problems of increasing reliability and ensuring the fault tolerance of the various classes and purposes of aircraft navigation systems. We present the results of studying failure detection methods to improve the accuracy and reliability of a strapdown functionally redundant inertial unit. The resulting structural redundancy of the strapdown inertial measurement unit is designed to increase the fault tolerance and accuracy of strapdown inertial navigation systems. The methods for detecting sensor failures in functionally redundant inertial units are based on the use of the equations of functionally redundant inertial unit compliance to nominal requirements for the accuracy of measuring the input action vector. To describe the methods for detecting and eliminating failed sensor and algorithm designs based on them, we gave the mathematical models of the measurement vector of functionally redundant inertial units concerning the measured vector and the error identification condition, including the residual of the matching equations with the size due to the level of redundancy, determining the total number of matching equations. The main criterion for determining a failed sensor is non-compliance with the nominal value of the residual included in a certain number of matching equations of the information received from such meters. The developed algorithms are examined using simulation methods. The study of the selected structure of the functionally redundant inertial units shows that the proposed approaches are efficient. Also, we manage to identify the main characteristics of the algorithms for detecting sensor failures that are structurally a part of the functionally redundant inertial units.

**Keywords:** functional redundancy; inertial measurement unit; angular velocity sensor; accelerometer; strapdown inertial navigation system

## 1. Introduction

The current requirements toward maintaining the operability of navigation systems of aircraft performing various tasks to improve reliability characteristics in terms of ensuring a certain level of fault tolerance due to possible malfunctions during operation limit the use of traditional navigation systems. The performance degradation of onboard navigation information measurers structurally included in aircraft onboard navigation systems (not to mention their failure) can significantly affect the navigation system's performance in terms of the required accuracy and availability of the navigation information. The use of information redundancy has traditionally been one of the ways to solve such problems.

The aircraft control systems of various classes and purposes, including unmanned aerial vehicles and small spacecrafts, must meet the requirements toward ensuring reliability and operability in the case of possible failures. One of the ways to increase reliability and ensure fault tolerance has traditionally been the use of information redundancy [1–5].

One of the essential advantages of strapdown inertial navigation systems (SINSs) is the possibility of increasing the reliability of measuring the input parameter vector at lower total

costs compared to traditional inertial navigation systems based on gyrostabilized platforms when the goal is achieved via the redundancy at the system level [4,5]. In the case of the strapdown version, one can solve this problem by using an excessive number of measuring elements as part of an inertial measurement unit (IMU) [6–8]. In this case, the IMU is called functionally redundant because it can perform its function using various combinations, for example, accelerometers measuring the components of the specific force vector and angular rate sensors (ARSs) that provide information about the motion parameters of a moving object relative to the center of mass. Also, both the hardware (the failure of the most sensitive element) and information (as a measure of the accuracy of the output information) reliability are monitored [5].

Redundancy refers to the difference between the number of primary information sensors used in the IMU and its number, the minimum necessary for setting a measuring basis that provides the measurements of the specific force vector, and the absolute angular velocity vector. In general, the measuring system can be non-orthogonal [4,9], while the optimality of the functionally redundant inertial unit (FRIU) technical solution is still achieved, for example, in terms of the weight and size and economic indicators.

Many researchers have studied methods of fault detection and isolation of inertial measurement systems with redundant sensors in their structure. The most common of them include such methods as generalized likelihood testing (GLT), which is not able to detect and isolate the faults of several autonomous inertial measurement units (SIMUs) along a certain axial direction. This is because, when solving the decoupling matrix, row vectors of the orthogonal projection matrix are used, which do not satisfy the conditions of linear correlation. The GLT method is used to detect a system fault, while the linear prediction method is used to estimate the inertial sensor values. A faulty inertial sensor signal can be isolated by comparing the calculated value with changes in the inertial sensor signal, and information about the fault is recorded for system recovery. The work [10] analyzes the disadvantages of the GLT method and proposes a methodology for constructing the decoupling matrix by selecting maximal linearly independent systems from an array of orthogonal projections and then orthogonalizing and combining the maximal uncorrelated set.

The article [11] proposes a method for detecting and isolating multiple faults based on a GLT and a linear prediction approach. The work [12] shows the approach in terms of the fault identification accuracy of tiny faults with an acceptable computational complexity.

There is also a known group of methods, such as singular value decomposition (SVD) [13], which use the null space component of faults with respect to a configuration matrix to obtain the geometric fault detection and isolation (FDI) technique. As well as SVD and GLT, the most frequent FDI techniques imply the use of an optimal parity vector [14–16] approach usually intended to recognize false alarms and wrong isolations.

One of the most recent studies [4] reviews the most common schemes of the FRIU configuration and its ensuring fault tolerance methods; a review of redundant inertial navigation technology can be found in [5].

Summarizing the analytical review, it can be concluded that in the reviewed works devoted to the consideration of ensuring FRIU-based fault tolerance methods, the effectiveness of a single method is usually considered in terms of the basic criteria for the effectiveness of its work, such as the time of failure detection [10–16]; its fault identification accuracy (i.e., depending on the FRIU working state [17]); the FRIU configuration and, related with this, the lever arm effects [18]; and the frequency of false alarms and wrong isolations, together with a study of general inertial system reliability [19] and its accuracy [20–23].

The research object is a fault-tolerant spacecraft SINS, configured with another one to be functionally and structurally redundant, so the research is quite specific, requiring special investigation and incurring problems that were not considered before. The methods of detecting failures to improve the accuracy and reliability of the strapdown FRIU are

considered. In this case, the arising structural redundancy of the strapdown IMU (SIMU) is intended to increase the SINS fault tolerance and accuracy.

This paper is organized as follows. In Section 2, the classification of the sensor failure types as well as the concept of a fault tolerance system design providing the two-level procedures with its mathematical models are given. The methodology of the failure detection algorithm operability confirmation, including several scenarios and simulation results, is presented in Section 3. A discussion of the simulation results, including a comparison to the results of other recent studies, is given in Section 4. Section 5 is devoted to the research conclusions.

## 2. Materials and Methods

This paper aims to analyze the possibility of using FRIUs to increase the reliability and ensure the fault tolerance of aircraft navigation systems of various classes and purposes. To achieve the research goal, we set the following tasks:

- Classify the types of sensor failures included in the FRIU;
- Develop a mathematical model for implementing algorithms for detecting and eliminating failed sensors in terms of the established classification;
- Design a research methodology for the proposed algorithms for detecting and eliminating failed sensors;
- Perform a simulation in accordance with the developed methodology of research on algorithms for detecting and eliminating failed sensors;
- Analyze the research results and estimate the operability of the proposed technical solutions and the main performance parameters of the developed algorithms.

### 2.1. Classification of Sensor Failure Types

The inertial measurement fault tolerance system is designed to detect a failure in the measurements of an IMU and identify a failed sensor. The failure of an inertial sensor means exceeding the estimate of the measurement error limits in the acceptable range of values.

Depending on the effect on the navigation parameter calculation error using the SINS algorithm, we suggest dividing the inertial sensors into three types:

- Instantaneous failure, which is when the measurement error estimate with the inertial sensor exceeds the limits of the tolerance range of values, allowing one to identify the failure in one measurement cycle unambiguously;
- Medium-level failure, which can be detected only after the accumulation and analysis of the statistical measurement indicators;
- Low-level failure, which is when using measurements of a failed inertial sensor leads to an increase in the errors of the SINS at a speed higher than that guaranteed by the passport characteristics.

The concept of a fault tolerance system design for inertial measurements based on two inertial information units identical in type and characteristics relies on a redundant number of measuring elements as part of a single (consisting of two inertial information units) IMU. In this case, the IMU is called functionally redundant because it can perform its function using various combinations of accelerometers that measure the components of the specific force vector and ARSs that provide information about the movement parameters of a moving object relative to the mass center.

Redundancy refers to the difference between the number of primary information sensors used in the IMU and their number, which is minimally required for creating a measuring basis that provides the specific force vector and the absolute angular velocity vector measurements.

The geometry of the FRIU measuring axes' location relative to each other is represented by two conditional groups in orthogonal and asymmetrically arranged directions. In the latter group, the sensors' measuring axes can be directed along the cone generatrix with different half-angles [2,3,7] or perpendicular to the regular polyhedra faces, depending on

the required redundancy level [9]. There are algorithms based on which failed sensors are detected and eliminated [24,25].

For sensors measuring the projections of the input action vector on their measuring axes (accelerometers and absolute angular velocity sensors), the non-redundant measuring basis (at a redundancy level equal to (in Equation (1)) $m = k - 3$, that is, at $k = 3$, $m = 0$) is designed using 3 sensors whose measuring axes are non-collinear and non-planar. The total number of different measurement bases in a unit of k sensors with a redundancy level of $m = k - 3$ is calculated using Equation (1):

$$n = C_k^3 = C_{m+3}^3,$$

(1)

and the total number of different measuring bases in a unit of 6 sensors reaches 20.

The concept of a fault tolerance system design provides for a three-level failure detection procedure:

- Failure detection in one measurement cycle (Level I procedure);
- Failure detection based on an analysis of the measurement residual statistical characteristics (Level II procedure);
- Failure detection based on complex information processing from the SINS built according to the FRIU measurements (Level III procedure).

We discuss the procedures of the first two levels and describe their main relationships in this section.

*2.2. Failure Detection in One Measurement Cycle*

As mentioned above, the total number of different measuring bases in a unit of 6 sensors reaches 20. This step allowed us to identify the failure of two sensors in one measurement cycle and detect the failure of the third. The basis for an algorithm for the Level I procedure design is the information about the attitude of the measuring axes of the sensors in the axes of the coordinate system (CS) "O" associated with the installation base on which both inertial information units are located. Such a characteristic of the IMU is called an alignment one, and for a unit of 6 sensors, it is presented as a direct cosine matrix (DCM) of the size (6 × 3).

Let the unit vector characterize the attitude of the measuring axis of the i-th sensor in the CS "O". Then, the transposed alignment matrix of a 6-sensor unit can be represented as follows (Equation (2)):

$$E_O^T = [\bar{e}_{o1}, \quad \ldots, \quad \bar{e}_{ok}],$$

(2)

where $E_O^T$—the transposed alignment matrix; $\bar{e}_{oi}$—the vectors of the direct cosines of the size (1 × 3) (i = 1 ... k), which make up the DCM of the size (i × 3), describing the attitude of the measuring axes of the *i* sensors in the CS "O", and the mathematical model of the FRIU measurement vector ($\overline{m}$) concerning the measured vector $\overline{V}_O$ takes the following form (Equation (3)):

$$\hat{m} = E_O \overline{V}_O + \Delta \overline{m},$$

(3)

where $\hat{m}$—the measured value of the vector components of the FRIU; $E_O$—the alignment matrix of the 6-sensor unit; $\overline{V}_O$—the measured vector, $\Delta \overline{m}$—a vector of the size (6 × 3) of the sensor instrumental errors.

Using the normal distribution law of the unit sensor instrumental errors (typical for measuring elements made using unified technology), the estimation of the maximum likelihood of the measured vector is achieved using the weighted least squares algorithm (Equations (4)–(6)):

$$\hat{V} = H\hat{m},$$

(4)

$$\hat{V} = (E_O^T Q E_O)^{-1} E_O^T Q,$$

(5)

$$Q = K_m^{-1},$$

(6)

where $\hat{V}$—the estimated vector; $H$—the transformation matrix of the size ($3 \times 6$) of the measurement vector of the single-component unit sensors ($\overline{m}$) into the measured vector estimate; $\hat{m}$—the FRIU sensor measurement vector of the size ($6 \times 1$); $E_O$—the direct cosine matrix (alignment) of the size ($6 \times 3$), describing the attitude of the measuring axes of the sensors in the CS "O"; and $K_m$—the measurement error covariance matrix of the size ($6 \times 6$).

The algorithm (Equations (4)–(6)) provides the smallest variance of the vector $\overline{V}$ error estimate on $\left(\sigma_v^2\right)$ in the asymptotic sense and the finite domain.

The measured vector in the expression (Equation (2)) can be the specific force vector ($\overline{V} = \overline{n}$) or the absolute angular velocity vector ($\overline{V} = \overline{\omega}$).

With equally accurate measurements, when the weight matrix is the identity one, the optimal algorithm is transformed into the least squares method (LSM) algorithm and reduced to multiplying the vector $\overline{m}$ by a constant pseudo-inverted matrix, the use of which is justified in the general case, with the lack of reliable information about the statistical characteristics of the measuring element errors:

$$H = \left(E_O^T E_O\right)^{-1} E_O^T, \tag{7}$$

An algorithm to be designed for detecting and identifying failures in one measurement cycle should use the IMU matching equations of the input action vector's measurement accuracy nominal requirements.

Let $\overline{\rho}$ be the s-dimensional residual matching equation vector (the dimension of the vector is due to the redundancy level that determines the matching equation number), and let $\overline{\varepsilon}_{tol}$ be these s-dimensional residual tolerance values' vector. Then, the conditions for the IMU error identification exceeding the nominal values can be expressed by the following vector equation:

$$\overline{\rho} = [C]\hat{m} \le \overline{\varepsilon}_{tol}, \tag{8}$$

where $\overline{\rho}$—the s-dimensional residual matching equations, s—the number of matching equations, $[C]$—the matching matrix of the size (s $\times$ k) corresponding to the condition (Equation (9)), $\hat{m}$—the k-dimensional unit sensor real measurement vector, and $\overline{\varepsilon}_{tol}$—the s-dimensional vector of these residual tolerance values.

$$[C]E_O = 0, \tag{9}$$

When the condition (Equation (9)) is met, the vector $\overline{\varepsilon}_{tol}$ can be found from the expression:

$$[C]\Delta\overline{m} \le \overline{\varepsilon}_{tol}, \tag{10}$$

The validity of this expression can be established by substituting (Equation (4)) into the main theoretical expression (Equation (8)). Indeed, when the condition (Equation (9)) is met, the useful signal is excluded from the expression (Equation (8)), and only the residual due to the measurement error remains.

To find the $[C]$ matrix elements, a system of algebraic equations corresponding to the measurements of four sensors should be used. Using the condition (Equation (9)) for a unit of 6 sensors, 15 equations have to be made, each responsible only for the sensors that the signals are part of.

With known values of the $[C]$ matrix elements for the 6-sensor IMU, 15 matching equations can be formed, each of which includes the data of only 4 measuring elements ($m_i$):

$$
\begin{aligned}
C_{1.1}m_1 + C_{1.2}m_2 + C_{1.3}m_3 + C_{1.4}m_4 &= \rho_1 \le \overline{\varepsilon}_{1tol}; \\
C_{2.1}m_1 + C_{2.2}m_2 + C_{2.3}m_3 + C_{2.4}m_4 &= \rho_2 \le \overline{\varepsilon}_{2tol}; \\
C_{3.1}m_1 + C_{3.2}m_2 + C_{3.3}m_3 + C_{3.6}m_6 &= \rho_3 \le \overline{\varepsilon}_{3tol}; \\
&\dots \\
C_{15.3}m_3 + C_{15.4}m_4 + C_{15.5}m_5 + C_{15.6}m_6 &= \rho_{15} \le \overline{\varepsilon}_{15tol}.
\end{aligned} \tag{11}
$$

The information from each measurer is included in 10 specific equations, which, in case of failure, do not correspond to the nominal value of the residual. The failed sensors are determined by the number of these equations.

When establishing the tolerance for the residual of the matching equations, we assumed that the instrumental errors of the SINS's measuring elements made using unified technology are independent and normally distributed. Then, according to Equations (8)–(10), the residual vector component SD can be calculated as follows:

$$\sigma_{\varepsilon i_{tol}} = \sigma_{\Delta m_{\max}} (\sum_{j=1}^{n} C_{ij}^2)^{1/2},$$
(12)

where $\sigma_{\varepsilon i_{tol}}$—the residual vector component SD, and $\sigma_{\Delta m_{\max}}$—the unit first measurement norm SD of $i = 1, \ldots, 15$; $j = 1, \ldots, 6$.

In (Equation (12)), the $\Delta m_{\max}$ first norm of the measurement vector of the unit is calculated as the maximum modulo value of the error component of the measurement vector at the time of calculating the residual vector.

However, the residual vector determination, which ensures the specified navigation accuracy and reliable identification of failed sensors, is a responsible engineering task, which should be solved by using reasoned simulation and hardware-in-the-loop simulation.

### 2.3. Failure Detection Based on the Analysis of Measurement Residual Statistical Characteristics

To detect faulty measurements caused, for example, by the failure of inertial sensors (in the form of an abrupt change in readings or an increasing error (medium and low-level failures)), an algorithm to test the null hypothesis about the non-bias of the matching equation residuals should be used, an alternative to which is the hypothesis about the presence of the average value in the residuals above the thresholds, set according to the manufacturers of inertial sensors and confirmed by the results of experiments (simulation and hardware-in-the-loop simulation). The hypothesis that the mean values of the differences are equal to zero is tested based on the normalized difference criterion (Student's criterion [26]) between the actually obtained statistical characteristics and the predicted ones.

To calculate the statistical characteristics of the measurements based on the last $l$ cycles, one should calculate the estimates of the moving averages of the residuals of the matching equations $M_{\rho_i}(t_k)$ and the variances of the $D_{\rho_i}(t_k)$ residuals for each correspondence equation $i$ at the time $t_k$:

$$M_{\rho_i}(t_k) = (\sum_{j=1}^{n} \rho_i(t_j))/l,$$
(13)

$$D_{\rho_i}(t_k) = (\sum_{j=1}^{n} \rho_i(t_j) - M_{\rho_i}(t_k))^2)/l,$$
(14)

where $M_{\rho_i}(t_k)$—the matching equations on the $l$ cycles residual estimates, $\rho_i(t_j)$—the $i$-th residual of the matching equations at the time $t_j$, and $l$—the number of cycles used to calculate the moving average.

The hypothesis of the non-bias of the residuals of the matching equations at a time $t_k$ should be tested as follows:

$$q_{\rho_i}(t_k) = (M_{\rho_i}(t_k) - M_{i\rho})/Y_{\rho_i}(t_k),$$
(15)

where $q_{\rho_i}(t_k)$—the calculated value of the criterion for testing the hypothesis, $Y_{\rho_i}(t_k) = (D_{\rho_i}(t_k) + D_{i\rho}/d)^{1/2}$, where $M_{i\rho}$ and $D_{i\rho}$ are determined according to the data from the inertial information unit manufacturer.

The condition for making a decision on the presence of errors above the threshold set by the manufacturer in the measurements of the sensors included in the $i$-th characteristic equation, with $r$ the number of the freedom level, has the following form:

If $q_{\rho_i}(t_k) > q_{crit}$, then the hypothesis is rejected: the presence of faulty measurements in the characteristic equation are diagnosed. The critical value of the criterion $q_{crit}$ is selected based on the condition $J(q_{crit}) = 1 - \alpha_r$. $\alpha_r$—the significance level of the criterion (probability of a type-1 error), and $J$—the Laplace function.

Similar to Section 2.2, the failed sensor should be determined according to the matching equation numbers.

It is advisable to organize fast ($n \approx 10$), medium ($n \approx 30$), and slow cycles ($n \approx 60$ s accordingly) for testing the hypothesis of the non-bias of the residuals, which will ensure the rapid detection of failures not detected by the Level I procedure and, at the same time, detect medium-level failures.

## 3. Results

To analyze the algorithms for detecting and eliminating failures of FRIU sensors, we provided an analytical assessment of the fault tolerance system performance in terms of identifying the possibilities for detecting and identifying failures with a different number of simultaneously failed inertial sensors.

The Level I and Level II procedures of failure detection and identification are based on the use of the so-called matching equations (Equation (10)). In the case of using two inertial measurement units for the FRIU design, the number of such equations is equal to 15.

If there is a failure of one sensor for 10 matching equations, the condition will not be met (see Equation (8)), and the hypothesis will not be confirmed (see Equation (14)); however, for the 5 remaining equations, this condition will be met and, respectively, the hypothesis will be confirmed. If there is a failure of two sensors for the 14 matching equations, the considered condition will not be met, and the hypothesis will also not be confirmed. At the same time, for the one remaining equation, the condition will be met, and the hypothesis will be confirmed. Finally, if there are failures of three or more sensors for all 15 matching equations, the verification condition will not be met, and the hypothesis will not be confirmed.

In this regard, it can be concluded that the Level I and Level II procedures make it possible to identify the failure of two sensors and detect the failure of the third one.

To verify the operability of the considered algorithms for ensuring the fault tolerance of the SINS measurements with an excessive number of measurers and identify their main characteristics, we performed a full-fledged simulation, which involved considering the functionality of the proposed system for ensuring the fault tolerance of the FRIU measurements in various scenarios. The key features of such scenarios can be divided into several groups:

- The study of the sensitivity of the considered system for ensuring the fault tolerance of SINS measurements with an excessive number of meters at different levels of sensor accuracy (systematic and noise components) of the FRIU (Level I and Level II procedures);
- An assessment of the requirements for the accuracy of the FRIU alignment based on two inertial information units;
- An assessment of the influence of the relative angular oscillations of two inertial information units.

### 3.1. Sensitivity Study with the Use of Various Accuracy Classes' Sensors

In the first group of scenarios, we tried to identify the dependence of the main parameters of the algorithm for detecting sensor failure (residuals of the matching equations and their standard deviation (SD) from the values of the main errors of the sensors under study), with various levels of systematic and noise components in the range from $10^{-3}$ to 10 deg/hour for the ARSs and from $10^{-5}$g to $10^{-1}$g m/s$^2$ for the accelerometers for the same trajectory conditions in which the FRIU operates. Table 1 provides examples of the calculations of the tolerance values of the residuals of the matching equations and their SD systems when operating algorithms of the Level I procedure (Figures 1 and 2 clearly

display it). Tables 2 and 3 display an example of calculating the tolerance values of the residuals of the matching equations according to the Student's criterion for fast, medium, and slow cycles (Level II procedure) (Figures 3 and 4).

**Table 1.** Acceptable values of residuals and their SD matching equations (ARSs) (Level I procedure).

| ARSs Systematic Error Value, deg/Hour | | | | | |
| --- | --- | --- | --- | --- | --- |
| 10 | | $10^{-1}$ | | $10^{-3}$ | |
| $\varepsilon_{tol}$ | $\sigma\varepsilon_{tol}$ | $\varepsilon_{tol}$ | $\sigma\varepsilon_{tol}$ | $\varepsilon_{tol}$ | $\sigma\varepsilon_{tol}$ |
| 1.89 | 4.01 | 0.21 | 0.45 | $1.76 \times 10^{-3}$ | $3.74 \times 10^{-3}$ |
| Accelerometers' Zero Drift Value, m/s$^2$ | | | | | |
| $10^{-1}$ g | | $10^{-3}$ g | | $10^{-5}$ g | |
| $\varepsilon_{tol}$ | $\sigma\varepsilon_{tol}$ | $\varepsilon_{tol}$ | $\sigma\varepsilon_{tol}$ | $\varepsilon_{tol}$ | $\sigma\varepsilon_{tol}$ |
| 0.37 | 0.78 | 0.06 | 0.35 | 0.04 | 0.18 |

**Table 2.** Acceptable values of residuals and their SD matching equations (ARSs) (Level II procedure).

| ARSs Systematic Error Value, deg/Hour | | | | | |
| --- | --- | --- | --- | --- | --- |
| 10 | | $10^{-1}$ | | $10^{-3}$ | |
| $\varepsilon_{tol}$ | $\sigma\varepsilon_{tol}$ | $\varepsilon_{tol}$ | $\sigma\varepsilon_{tol}$ | $\varepsilon_{tol}$ | $\sigma\varepsilon_{tol}$ |
| "fast" cycle ($n = 10$) | | | | | |
| 11.61 | 37.75 | 0.115 | 0.374 | $1.26 \times 10^{-3}$ | $4.1 \times 10^{-3}$ |
| "average" cycle ($n = 30$) | | | | | |
| 2.166 | 5.956 | $2.1 \times 10^{-2}$ | $5.9 \times 10^{-2}$ | $2.4 \times 10^{-4}$ | $6.5 \times 10^{-4}$ |
| "slow" cycle ($n = 60$) | | | | | |
| 0.7323 | 1.887 | $7.3 \times 10^{-3}$ | $1.87 \times 10^{-2}$ | $2.1 \times 10^{-4}$ | $2.1 \times 10^{-4}$ |

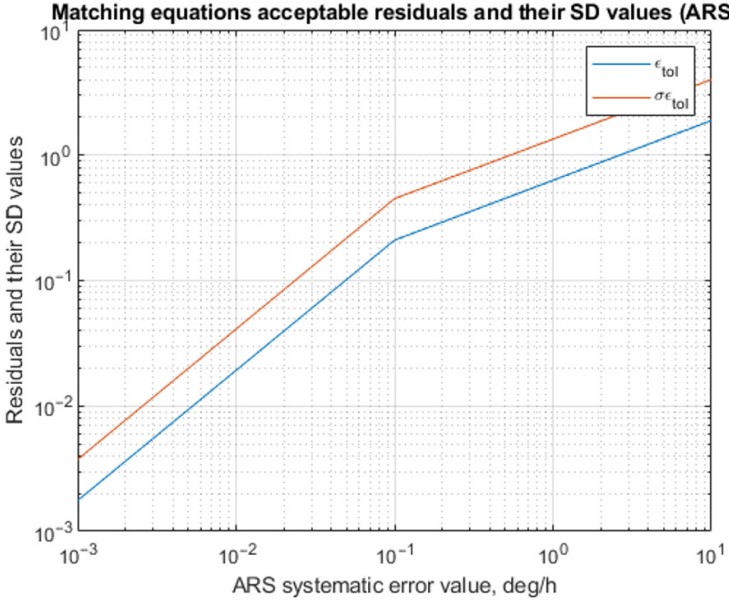

**Figure 1.** Matching equations' acceptable residuals and their SD (for ARSs) (Level I procedure).

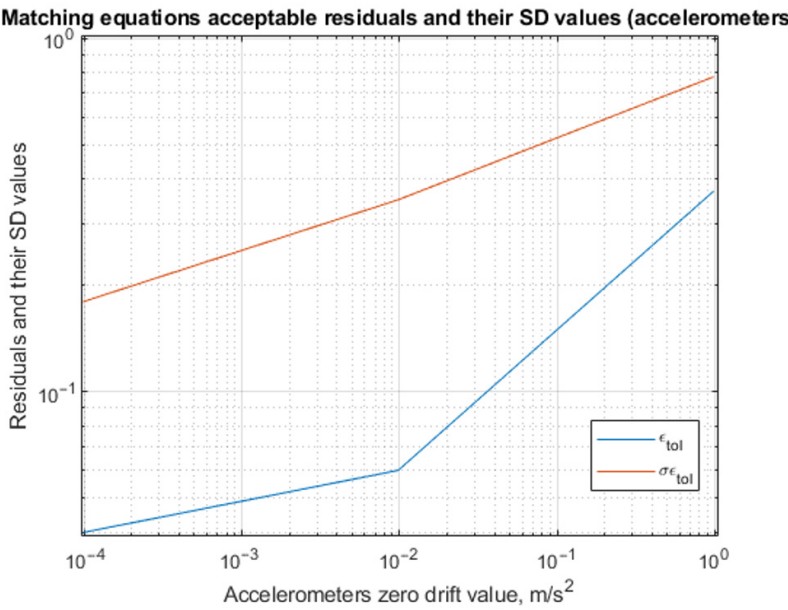

**Figure 2.** Matching equations' acceptable residuals and their SD (for accelerometers) (Level I procedure).

**Table 3.** Acceptable values of residuals and their SD matching equations (accelerometers) (Level II procedure).

| Accelerometers' Zero Drift Value, m/s$^2$ | | | | | |
|---|---|---|---|---|---|
| $10^{-1}$g | | $10^{-3}$g | | $10^{-5}$g | |
| $\varepsilon_{tol}$ | $\sigma\varepsilon_{tol}$ | $\varepsilon_{tol}$ | $\sigma\varepsilon_{tol}$ | $\varepsilon_{tol}$ | $\sigma\varepsilon_{tol}$ |
| "fast" cycle ($n = 10$) | | | | | |
| 1.139 | 3.7 | 0.01 | 0.033 | $1.12 \times 10^{-3}$ | $3.65 \times 10^{-3}$ |
| "average" cycle ($n = 30$) | | | | | |
| 0.212 | 0.584 | 0.002 | $5.3 \times 10^{-3}$ | $1.96 \times 10^{-4}$ | $5.43 \times 10^{-4}$ |
| "slow" cycle ($n = 60$) | | | | | |
| $7.18 \times 10^{-2}$ | 0.185 | $6.7 \times 10^{-4}$ | $1.6 \times 10^{-3}$ | $6.16 \times 10^{-5}$ | $1.61 \times 10^{-4}$ |

The results obtained within the framework of this study allow for identifying the levels of sensitivity of the proposed approach. For the Level I procedure, for example, the ARS threshold values and their SD of the matching equations almost fit the linear function (see Figure 1) of its accuracy class, and it is the most sensitive for the consumer grade (10 deg/hour) and equals approximately 20% for its mean value. At the same time, the threshold levels for the tactical ($10^{-1}$ deg/hour) and navigation ($10^{-3}$ deg/hour ARS drift rate, respectively (i.e., IMU 1000 [27])), on average, are two times bigger than its accuracy level. The accelerometer threshold values and their SD of the matching equations also almost fit the linear function (see Figure 2) of its accuracy class and are also the most sensitive for the consumer grade ($10^{-1}$g m/s$^2$) and equal approximately 37% for its mean value. At the same time, the threshold level for tactical ($10^{-3}$g m/s$^2$) is six times and navigation ($10^{-5}$g m/s$^2$ accelerometer zero drifts, respectively) is two orders of magnitude greater than its accuracy level.

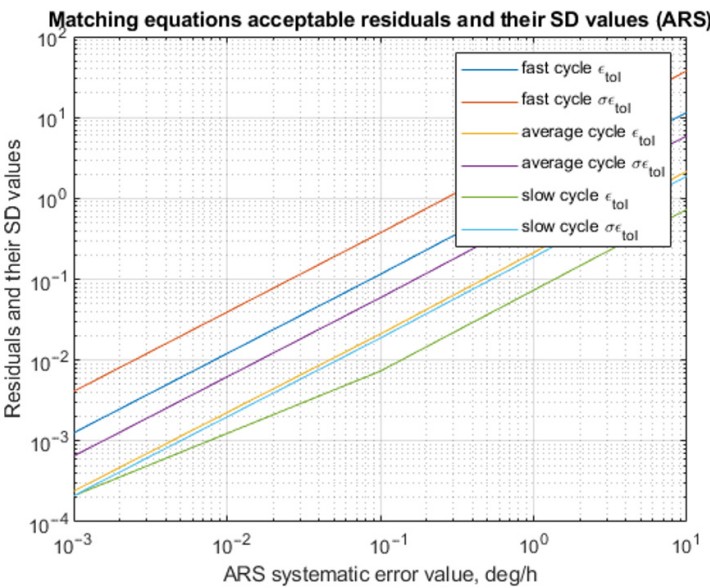

**Figure 3.** Matching equations' acceptable residuals and their SD (for ARSs) (Level II procedure).

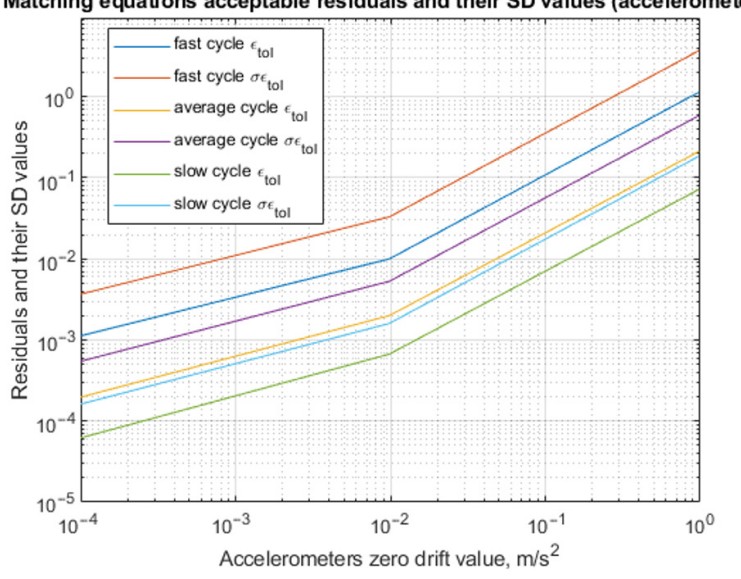

**Figure 4.** Matching equations' acceptable residuals and their SD (for accelerometers) (Level II procedure).

For the Level II procedure, the ARS threshold values and their SD of the matching equations have an even more linear function fit for the accuracy class, compared to the Level I procedure (see Figure 3), and the sensitivity increases within the sensor accuracy class logarithmically from the "fast" to "slow" cycle simulation test conditions. For the consumer grade (10 deg/hour), the threshold levels have the range of 116% for "fast", 21% for "medium", and 7% for "slow" cycles, respectively; at the same time, for the tactical grade ($10^{-1}$ deg/hour), it has the range of 1115% for "fast", 210% for "medium", and 73% for "slow" cycles, respectively. Last, the threshold levels for the navigation grade ($10^{-3}$ deg/hour ARS drift rate, respectively) are in the range of 126% for "fast", 24% for "medium", and 21% for "slow" cycles, respectively. The accelerometer threshold values obtained within the Level II procedure simulation and their SD of the matching equations also almost fit the linear function (see Figure 4) of its accuracy class, and as well as for the ARS, its sensitivity increases within the sensor accuracy class logarithmically from the "fast" to "slow" cycle simulation test conditions. For the consumer grade ($10^{-1}$g m/s$^2$),

the threshold levels have the range of 116% for "fast", 21% for "medium", and 7% for "slow" cycles, respectively; at the same time, for the tactical grade ($10^{-3}$g m/s$^2$), it has the range of 101% for "fast", 20% for "medium", and 6.8% for "slow" cycles, respectively. Last, the threshold levels for the navigation grade ($10^{-5}$g m/s$^2$ accelerometer zero drifts, respectively) are in the range of 1141% for "fast", 199% for "medium", and 62% for "slow" cycles, respectively.

The obtained results allow for concluding that the combination of various Level II procedure cycles can lead to a preferred FDI accuracy level and can be customized and optimized due to the overall required system reliability level.

### 3.2. Assessment of Adjustment Accuracy Requirements

The realization of the potential advantages of functionally redundant meters is associated with the complexity of the measuring element unit (MEU) design. Generally, this circumstance leads to technical difficulties in ensuring the alignment characteristics of the unit and requires a reasonable assignment of tolerances to ensure the nominal attitude of its measuring axes.

In this regard, in the second group of the simulation modeling, the requirements for the accuracy of the MEU adjustment were evaluated, and the unit output information accuracy was investigated.

For a class of functionally redundant accelerometer units (AUs) of a conical structure with a nominal alignment characteristic of the form (Equation (16)):

$$H = [\sin\chi \cdot \cos\beta_i, \cos\chi, \sin\chi \cdot \sin\beta_i]^T, \tag{16}$$

where $H$—the transformation matrix, and where $i = 1, \ldots, k$, $\beta_i = 2\pi(i-1)/k$, the expression for the functional transformation matrix will take the form (Equation (17)):

$$H = \begin{bmatrix} \frac{2\cos\beta_1}{k\sin\chi}, & \cdots, & \frac{2\cos\beta_1}{k\sin\chi}, & \cdots, & \frac{2\cos\beta_k}{k\sin\chi} \\ \frac{1}{k\cos\chi}, & \cdots, & \frac{1}{k\cos\chi}, & \cdots, & \frac{1}{k\cos\chi} \\ \frac{-2\sin\beta_1}{k\sin\chi}, & \cdots, & \frac{-2\sin\beta_i}{k\sin\chi}, & \cdots, & \frac{-2\sin\beta_k}{k\sin\chi} \end{bmatrix}, \tag{17}$$

where $\chi$—the half-angle of the cone, along the generatrix of which the measuring axes of the sensors are uniformly located (with the $\beta_i$ angles).

For the MEU design chosen within the modeling, the first norm of the $H$ transformation matrix is determined by the second row:

$$\|H\|_I = \frac{1}{|\cos\chi|}, \tag{18}$$

The value of the angle $\chi$ in the selected configuration is 54.74° [4,9], which determines the values of the expression's corresponding norms. This step allowed us to estimate the maximum error of the MEU output depending on the adjustment errors, configuration of the unit, and flight conditions:

$$\left\|\delta\hat{V}\right\|_I = 2\|H\|_I \cdot \|\bar{r}_i\|_I \cdot \|\hat{V}\|_I |\Theta_{ij}|_{\max}, \tag{19}$$

$\|H\|_I = 1.7013$, $\|\bar{r}_i\|_I = 0.89$, ($\|\hat{V}\|_I = 1$ g–cruise flight mode).

As a result, the following calculated ratios can be obtained: $|\Theta_{ij}|_{\max} = |\delta n_i|/2.75$.

As part of the simulation, the requirements for the accuracy of the alignment were studied. The results from calculating the tolerance value of the adjustment error ($3\sigma$) are shown in Table 4. During the calculation, we assumed that the error of the AU output information ($|\delta n_i|$) should not exceed the accelerometer zero-offset error. Table 4 also shows the results for the orthogonal characteristics of the MEU structure and the conical structure with an angle $\chi$ equal to 78°.

**Table 4.** Identification of failed measuring elements in an IMU of six sensors.

| Characteristic of the Unit Structure | Tolerance Value of the Adjustment Error, ang. s. | | | |
|---|---|---|---|---|
| | $\delta n_i = 1 \times 10^{-4}\,g$ | $\delta n_i = 3 \times 10^{-4}\,g$ | $\delta n_i = 6 \times 10^{-4}\,g$ | $\delta n_i = 8 \times 10^{-4}\,g$ |
| Orthogonal, coinciding with the orts of the body frame | 10 | 30 | 60 | 80 |
| Conical $\chi = 54.74°$ | 7.3 | 22 | 44 | 58 |
| Conical $\chi = 78°$ | 2 | 6.4 | 13 | 17 |

The results obtained within the framework of the FRIU accuracy alignment requirements study based on two inertial information units shows that it depends on its structure and increases significantly better (up to 27% for conical ($\chi = 54.74°$) and 80% for conical ($\chi = 78°$)) as well as for the precise adjustment (10 ang. s. error) and for the rough one (80 ang. s. error) and allows for stating the fact of more strict requirements for the more non-orthogonal unit structure.

*3.3. Evaluation of the Relative Angular Oscillation Effect*

To ensure the design of the FRIU based on already existing orthogonal IMUs, providing the necessary technological design of their relative angular position, the developers of the FRIU should estimate the effect of angular fluctuations of the IMUs that will lead to a change in the values of the matrix elements $E_O$ (Equation (2)). The analytical expression for calculating the matching equations' residual error can be obtained by formally varying the expression (Equation (20))

$$\bar{\rho} = [C]E_O V_O + [C]\Delta m, \tag{20}$$

which can be obtained by substituting (Equation (3)) with (Equation (8)) in the resulting expression

$$\partial\rho = [C]\partial E_O \bar{V}_O \tag{21}$$

$$\partial E_O = [\delta\bar{e}_{O1}\dots,\delta\bar{e}_{Oi}]^T, \tag{22}$$

where $\partial E_O$—the matrix of the size ($6 \times 3$) of the unit measuring axes' adjustment errors.

The row of the matrix $\partial E_O$ describes the measuring axis adjustment error of the i-th sensor; it is expressed in terms of the vector product components $\delta\bar{e}_{Oi} = \bar{\Theta}_i \times \bar{e}_{Oi}$, one of the multipliers of which is the small rotation vector ($\bar{\Theta}_i$), which characterizes the difference between the real measuring axis attitude of the i-th sensor from the nominal one caused by angular fluctuations of two IMUs in their own housing.

The error in calculating the residual of the matching equations caused by the relative angular fluctuations of the two IMUs can be estimated using a simulation, presenting a model of the change in the components of the small rotation vector ($\bar{\Theta}_i$), for example, in the form of a harmonic oscillation with specified parameters (the circular oscillation frequency of 100 rad/s; the oscillation amplitude of $1°$; and the number of cycles–$10^5$, respectively).

Table 5 shows the results of the simulation with the ARSs as the measuring elements used.

The results obtained within the framework of the relative angular oscillation effect study shows the level of its contribution to the overall matching equation residual calculating error. Depending on the specific construction and increased dynamics, this issue has the level up to percent fractions in its mean value but adds considerable noise, which increases the probability of type-1 errors.

**Table 5.** Estimation of the matching equations' residual calculating error for ARSs (deg./hour).

| $\theta_{max}$, deg. | $V_O$, deg/Hour | | | | | |
|---|---|---|---|---|---|---|
| | **5** | | **10** | | **20** | |
| | **ME \*** | **SD** | **ME** | **SD** | **ME** | **SD** |
| 1/360 | $10^{-6}$ | 0.44 | $10^{-6}$ | 0.88 | $10^{-6}$ | 1.7 |
| 1/60 | $10^{-5}$ | 2.6 | $10^{-5}$ | 5.2 | $10^{-5}$ | 10.5 |
| 1 | $10^{-4}$ | 158 | 0.001 | 316 | 0.002 | 633 |

\* ME—mathematical expectation (expected value).

Table 6 shows the results of the simulation with accelerometers as the measuring elements used.

**Table 6.** Estimation of the matching equations' residual calculating for accelerometers (m/s$^2$).

| $\theta_{max}$, deg. | $V_O$, m/s$^2$ | | | | | |
|---|---|---|---|---|---|---|
| | **9.81** | | **2 × 9.81** | | **3 × 9.81** | |
| | **ME \*** | **SD** | **ME** | **SD** | **ME** | **SD** |
| 1/360 | $10^{-9}$ | $10^{-4}$ | $10^{-9}$ | $10^{-4}$ | $10^{-9}$ | $10^{-4}$ |
| 1/60 | $10^{-9}$ | $10^{-3}$ | $10^{-8}$ | 0.02 | $10^{-8}$ | 0.004 |
| 1 | $10^{-7}$ | 0.08 | $10^{-7}$ | 0.17 | $10^{-6}$ | 0.25 |

\* ME—mathematical expectation (expected value).

The simulation showed that the proposed approach to detect failures of inertial sensors structurally included in the FRIU performs well. The approach considered is a set of functional algorithms that implement the concept of a system design that is ensuring the fault tolerance of the functioning of both the sensors and systems that operate using the measuring information received from such sensors, primarily SINSs. In addition, this approach makes it possible to detect failures in one measurement cycle and to identify failures based on the analysis of the statistical characteristics of the measurement residuals.

The study of the sensitivity of failure detection algorithms when using sensors of various accuracy classes in the first group of simulation scenarios made it possible to determine the tolerance values of residuals and their SD, as well as to establish thresholds for making a decision to exclude failed sensors from the SINS's functional algorithms. The given simulation results allowed us to select sensitive elements for the FRIU design for the configuration scheme under consideration with the selected redundancy characteristics.

The assessment of the requirements for the FRIU accuracy alignment based on two inertial information units in the second group of simulation scenarios shows that the requirements for the accuracy of the alignment depend on the structure of the AU and increase significantly as the non-orthogonality of the unit structure increases. In more dynamic flight modes, the alignment accuracy requirements may exceed the design and technological capabilities of their implementation. In this case, analytical alignment is the only way to solve the problem.

The third group of simulation scenarios made it possible to determine the assessment of the influence of the relative angular oscillations of the two inertial information units and to obtain numerical values of error estimates for calculating the residual of the matching equations for the selected model of the external disturbance of the considered FRIU.

## 4. Discussion

The analysis of the simulation results allowed us to conclude that the proposed approach to detecting failures of inertial sensors structurally included in the FRIU operates well. The numerical values obtained in Tables 1–3 reflect the dependences of the main parameters of the algorithm for detecting sensor failure, the residual of the matching

equations, and its SD on the values of the main errors of the sensors under study in various levels of the systematic and noise components of the ARSs and accelerometers. In this case, the tolerance values of the residuals of the matching equations can be obtained directly during the operation of the algorithm. The approach is somewhat universal because there is no need to work with a useful signal; it is enough to have an a priori assessment of the characteristics of the sensors and work on the residuals due to the error of its measurements.

A feature of the proposed approach that distinguishes it from those presented in other works [10–12,24,25] is the ability of the algorithm to detect so-called medium-level failures using the statistical hypothesis-testing technique. Among the other things, the algorithm provides the smallest variance of the vector error of estimation in the asymptotic sense and the finite domain. The way it is designed is that it works specifically with the residuals—information about all the types of failures is contained in it and a useful signal can be excluded from consideration, thereby avoiding errors affecting the occurrence of false alarms [21,22]. One of the advantages of the proposed approach is the high value of fault identification accuracy with a high probability due to the use of a large number of correspondence equations, which greatly simplifies the procedure for identifying failed sensors, eliminates false alarms with an appropriately pre-performed procedure for factory calibration of sensors, and, in addition, unlike the other works, is a customizable parameter [17,19,21,22].

Moreover, the algorithm takes into account the influence of tolerances on ensuring the nominal attitude of the FRIU measuring axes, which allows for solving the problems associated with alignment that inevitably arise, for example, when installing ready-made measuring units in a structure that provides a redundancy of measurements [17,19,22,23]. The assessment of the requirements for the accuracy of the alignment of the MEU (Table 4), with the parameter chosen as the main criterion (the accuracy of the output information of the unit), allowed us to establish the dependence of the structure of the MEU and the dynamics of the movement of the object on these requirements. In this case, analytical accounting of the adjustment parameters may be the only way to improve the accuracy of the meters.

Finally, due to the conditions of the numerical experiment, in contrast to published works in the subject area [3,7,10–12,17,19,22,23], the algorithm provides for the possibility of considering the influence of mutual angular oscillations of the FRIU based on already existing orthogonal IMUs, for which the necessary technological design of their mutual angular position is provided. The numerical values obtained during the experiment of the estimates of the errors in calculating the residual of the matching equations for the selected model of the external disturbance of the considered FRIU (Tables 5 and 6) make it possible to increase the final accuracy of the algorithm for detecting and eliminating sensor failures in the presence of such disturbances.

## 5. Conclusions

As a result of this study, we suggested ways to detect failures to improve the accuracy and reliability of the strapdown FRIU. In this case, the arising structural redundancy of the SIMU is intended to increase the SINS's fault tolerance and accuracy levels.

When substantiating the concept of fault tolerance, we classified the types of failures. Thus, we showed the concept of a fault tolerance system design that provides a three-level failure detection procedure to detect failures of all types.

The possibilities of detecting and identifying failures using the proposed approach were studied using simulation-modeling methods. We considered a wide range of modeling scenarios, providing the most comprehensive evaluation of the detecting sensor failure algorithms' main characteristics that are structurally an FRIU part.

The research results allowed us to conclude that the proposed approach to the fault tolerance system design is working. Also, an analytical assessment of the fault tolerance system characteristics in terms of an evaluation of the capabilities of detecting and identifying failures with a different number of simultaneously failed inertial sensors revealed

that Level I and Level II procedures allowed us to identify the failure of two sensors and to detect the failure of the third one.

The research novelty lies in the ability of the algorithm to detect medium-level failures with the help of the statistical hypothesis-testing technique, taking into account the influence of tolerances on ensuring the nominal attitude of the measuring axes of functionally redundant inertial units and the relative angular oscillation of the inertial measurement unit structurally included in the strapdown inertial navigation system.

Finally, the use of first- and second-level procedure sets makes it possible to provide the so-called FRIU predictive diagnostics and to warn the consumer in advance of the need to exclude such measurements from the navigation solution.

A further direction of the subject area can involve studying a Level III procedure based on the complex processing of information from parallel-executed SINS algorithms constructed from measurements of a functionally redundant measuring unit containing six identical ARSs and accelerometers included in two inertial information units. This procedure is assumed to allow for identifying and eliminating failures of low-level sensors.

**Author Contributions:** Conceptualization, K.K.V.; methodology, K.K.V.; software, M.V.Z. and A.N.P.; validation, I.M.K. and A.N.P.; formal analysis, I.M.K.; investigation, I.M.K.; resources, I.M.K.; data curation, A.N.P.; writing—original draft preparation, I.M.K.; writing—review and editing, I.M.K.; supervision, M.V.Z.; project administration, K.K.V.; funding acquisition, K.K.V. All authors have read and agreed to the published version of the manuscript.

**Funding:** The research was carried out within the state assignment of the Ministry of Science and Higher Education of the Russian Federation (theme no. FSFF-2023-0005).

**Data Availability Statement:** Data are contained within the article.

**Conflicts of Interest:** The authors declare no conflict of interest.

## Abbreviations

The following abbreviations are used in this manuscript:

| | |
|---|---|
| ARS | Angular Rate Sensors |
| AU | Accelerometer Unit |
| CS | Coordinate System |
| DCM | Direct Cosine Matrix |
| FDI | Fault Detection and Isolation |
| FRIU | Functionally Redundant Inertial Unit |
| GLT | Generalized Likelihood Testing |
| IMU | Inertial Measurement Unit |
| LSM | Least Squares Method |
| ME | Mathematical Expectation |
| MEU | Measuring Element Unit |
| SD | Standard Deviation |
| SIMU | Strapdown Inertial Measurement Unit |
| SINS | Strapdown Inertial Navigation System |
| SVD | Singular Value Decomposition |
| **Notation** | **Description** |
| k, l, m, n, s | Number of sensors, number of cycles used to calculate the moving average, redundancy level, total number of different measuring bases, number of matching equations |
| $E_O$ | Alignment DCM |
| $\bar{e}_{oi}$ | Vectors of the direct cosines of the size ($1 \times 3$), ($i = 1 \ldots k$) |
| $\hat{m}$ | Measured value of the FRIU vector components of the size ($6 \times 1$) ($\text{rad}/\text{m}/\text{s}^2$) |
| $\overline{V}_O$ | Measured vector of the size ($6 \times 1$) ($\text{rad}/\text{m}/\text{s}^2$) |
| $\Delta \overline{m}$ | Vector of the instrumental errors of the sensors ($\text{rad}/\text{m}/\text{s}^2$) |
| $\hat{V}$ | Estimated measurement vector of the size ($3 \times 1$) ($\text{rad}/\text{m}/\text{s}^2$) |
| H | Transformation matrix of the size ($3 \times 6$) |

| | |
|---|---|
| $K_m$ | The measurement error covariance matrix of the size ($6 \times 6$) |
| $\sigma_v^2$ | Measurement vector variance ($\text{rad}^2/\text{m}^2/\text{s}^4$) |
| $\bar{n}$ | The specific force vector of the size ($3 \times 1$) ($\text{m/s}^2$) |
| $\bar{\omega}$ | The absolute angular velocity vector of the size ($3 \times 1$) (rad) |
| $\bar{\rho}$ | The vector of the residuals of the matching equations |
| $[C]$ | The correspondence matrix of the size (s $\times$ k) |
| $\bar{\varepsilon}_{tol}$ | s-dimensional vector of residual tolerance values |
| $\sigma_{\varepsilon i_{tol}}$ | SD of the residual vector components |
| $\sigma_{\Delta m_{\max}}$ | SD of the measurement unit first measurement norm |
| $M_{\rho_i}(t_k)$ | Moving averages of the matching equation residuals at the time $t_k$ |
| $\rho_i(t_j)$ | $i$-th residual of the matching equations at the time $t_j$, within the cycle of moving average calculation (from $j = 1 \ldots l$) |
| $q_{\rho_i}(t_k)$ | Calculated value of the criterion for testing the hypothesis |
| $D_{i\rho}$ | Matching equation residual variance |
| $q_{crit}$ | The critical value of the criterion for testing the hypothesis |
| $J$ | Laplace function |
| $\alpha_r$ | Level of significance of the criterion (probability of type-1 error) |
| $\beta_i$ | FRIU configuration coordinate angles (rad) |
| $\chi$ | Cone half-angle (sensor measurement axis oriented along its generatrix) (rad) |
| $\partial \rho$ | The error vector of the matching equation residuals |
| $\partial E_O$ | Measuring axis adjustment error matrix |
| $\delta \bar{e}_{Oi}$ | Adjustment error direct cosine vector of the size ($1 \times 3$), (i = 1 $\ldots$ k) |
| $\overline{\Theta}_i$ | Small rotation vector (rad) |

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
