# Peer review of "Using Functionally Redundant Inertial Measurement Units to Increase Reliability and Ensure Fault Tolerance"

_inventions, doi:10.3390/inventions8060159_

Round 1

Reviewer 1 Report

Comments and Suggestions for Authors

The study explores the practicality of utilizing functionally redundant inertial units to address reliability and fault tolerance issues in the navigation systems of diverse aircraft types. The authors focus on improving the precision and dependability of strapdown functionally redundant inertial units through a thorough investigation into failure detection methods. The resulting structural redundancy in the strapdown inertial measurement unit is designed to elevate the fault tolerance and precision of strapdown inertial navigation systems. The study delves into the use of equations governing compliance with accuracy requirements to develop methods for detecting and rectifying sensor failures in functionally redundant inertial units. Mathematical models are presented to describe the measurement vector of functionally redundant inertial units concerning the measured vector and error identification conditions. The models include the residual of matching equations, influenced by the level of redundancy and determining the overall number of matching equations. The principal criterion for identifying failed sensors involves their deviation from the nominal value of the residual within a specific number of matching equations derived from sensor information. The authors developed algorithms for detecting sensor failures, which were then evaluated using simulation methods to ensure their effectiveness. The examination of the selected structure of functionally redundant inertial units illustrates the efficacy of the proposed approaches. he authors outline the key characteristics of the algorithms designed for detecting sensor failures, emphasizing their integration into the structure of functionally redundant inertial units. The manuscript demonstrates value. Nevertheless, certain aspects require attention and resolution prior to acceptance.

1.      The originality of the manuscript is uncertain. A limited number of references were examined, with the majority dating back 10-20 years. It is crucial to critically review recent literature to establish the necessity of this research.

2.      In lines 92-107 of the Materials and Methods section, the content appears redundant and lacks coherence. Kindly eliminate these statements.

3.      Kindly furnish the complete forms for all abbreviations used, as some are currently lacking their full forms.

4.      Numerous equations incorporating a multitude of symbols were employed. It is imperative to elucidate the meanings of these symbols, as comprehending the derivation of the equations proves challenging without such explanations. Additionally, incorporating a nomenclature section would be beneficial.

Comments on the Quality of English Language

The grammar appears to be correct; however, the language style is challenging to comprehend.

Author Response

A file with answers is attached.

Reviewer 2 Report

Comments and Suggestions for Authors

The article, submitted for review, discusses the evaluation of the feasibility of using functionally redundant inertial units to solve the problems of increasing the reliability and ensuring the fault tolerance of navigation systems of aircraft of different classes and purposes.  The article has several shortcomings such as the following
1`. The literature presented is incomplete and should be improved with more recent studies.
2, The mathematical considerations lack explanation of parameters ( not in all of them).
5, The article lacks IMU parameters and the type of aircraft used.
6, The results are sparsely discussed please correct this.

Author Response

A file with answers is attached.

Round 2

Reviewer 1 Report

Comments and Suggestions for Authors

The manuscript has improved significantly. It can be approved now.